# Selenium Status in Patients with Chronic Liver Disease: A Systematic Review and Meta-Analysis

**DOI:** 10.3390/nu14050952

**Published:** 2022-02-23

**Authors:** Yaduan Lin, Fanchen He, Shaoyan Lian, Binbin Xie, Ting Liu, Jiang He, Chaoqun Liu

**Affiliations:** 1Department of Nutrition, School of Medicine, Jinan University, Guangzhou 510632, China; yyw135@stu2016.jnu.edu.cn (Y.L.); lsy192735@163.com (S.L.); binbinxiejn@126.com (B.X.); lt1996jnu@163.com (T.L.); 2Institute of Land and Sea Transport Systems, Faculty of Mechanical Engineering and Transport Systems, Technical University of Berlin, 10623 Berlin, Germany; fanchen.he@campus.tu-berlin.de; 3Department of Mathematics and Physics, School of Biomedical Engineering, Southern Medical University, Guangzhou 510515, China

**Keywords:** selenium, chronic liver disease, fatty liver diseases, hepatitis, cirrhosis, liver cancer, selenium intake, meta-analysis

## Abstract

Background: The potential role of selenium in preventing chronic liver diseases remains controversial. This meta-analysis aimed to summarize the available evidence from observational studies and intervention trials that had evaluated the associations between body selenium status and chronic liver diseases. Methods: We comprehensively searched MEDLINE, Embase, Web of Science, and Cochrane Library from inception to April 2021. The study protocol was registered at PROSPERO (CRD42020210144). Relative risks (RR) for the highest versus the lowest level of selenium and standard mean differences (SMD) with 95% confidence intervals (CI) were pooled using random-effects models. Heterogeneity and publication bias were evaluated using the *I*^2^ statistic and Egger’s regression test, respectively. Results: There were 50 studies with 9875 cases and 12975 population controls in the final analysis. Patients with hepatitis (SMD = −1.78, 95% CI: −2.22 to −1.34), liver cirrhosis (SMD = −2.06, 95% CI: −2.48 to −1.63), and liver cancer (SMD = −2.71, 95% CI: −3.31 to −2.11) had significantly lower selenium levels than controls, whereas there was no significant difference in patients with fatty liver diseases (SMD = 1.06, 95% CI: −1.78 to 3.89). Moreover, the meta-analysis showed that a higher selenium level was significantly associated with a 41% decrease in the incidence of significant advanced chronic liver diseases (RR = 0.59, 95% CI: 0.49 to 0.72). Conclusion: Our meta-analysis suggested that both body selenium status and selenium intake were negatively associated with hepatitis, cirrhosis, and liver cancer. However, the associations for fatty liver diseases were conflicting and need to be established in prospective trials.

## 1. Introduction

Selenium has been recognized as an essential trace element in humans for decades, which exerts its biological functions in antioxidant defense, redox signaling, thyroid hormone metabolism, and immune response through various selenoproteins [1]. Currently, 25 selenoproteins have been identified in humans and most of them serve as oxidoreductases, with selenocysteine being the catalytic redox-active site. The intake of selenium varies greatly and ranges from deficient to toxic concentrations worldwide. With an insufficient intake of selenium in the body, the amount of selenoproteins decreases and can, therefore, affect the immune system by decreasing the development and functions of the thymus responsible for the production of macrophages and lymphocytes [1,2]. Extensive studies have indicated that selenium deficiency is an important contributing factor to the pathogenesis of numerous chronic diseases, such as cancer [3,4], cardiovascular diseases, diabetes, and liver diseases [5,6], as well as other disorders [3,7]. On the contrary, the excessive intake of selenium may cause oxidative damage, cytotoxicity, and increase DNA damage, often leading to nail fragility and hair loss [1,2].

Chronic liver diseases, including fatty liver diseases, hepatitis, fibrosis, cirrhosis, and cancer, are major global health burdens and account for approximately 2 million deaths per year worldwide [8]. The reason for liver damage is mainly related to extensive inflammation and oxidative stress, generated by excessive reactive oxygen species (ROS) production, which promotes liver diseases. Se deficiency induces a systematic redox imbalance and inflammation in the blood and causes pathological changes in the liver [9]. Due to the unique chemical reactivity of selenocysteines, several selenoproteins have been reported to mitigate and repair liver damage caused by ROS, including thioredoxin reductases (TXNRD), glutathione peroxidases 1 (GPX1), selenoproteins P (SELENOP), SELENOS, and SELENOK [10,11]. The normal range of selenium in serum of healthy individuals is 0.8–2 µmol/L (or 70–150 µg/L) [3,12]. Recently, substantial epidemiological studies have been performed to explore the link between selenium levels and chronic liver diseases risk. Plenty of studies have reported that, compared with healthy individuals, selenium levels in patients with chronic liver disease are lower when the selenium status is at the optimal level, especially in patients with advanced chronic liver disease such as hepatitis, cirrhosis, and liver cancer [13,14,15]. However, the results are not always consistent and its relationship with different severities of chronic liver diseases has been controversial. In the early stage of chronic liver diseases, some evidence linked lower blood selenium levels to fatty liver diseases compared to controls [5,16], while other studies found no or even a positive association of selenium levels and non-alcoholic fatty liver disease (NAFLD) [17]. In addition, some of the studies have investigated the association between selenium intake and the risk of chronic liver diseases but came out with inconsistent results. Furthermore, studies implied that maintaining an adequate amount of body selenium or selenium supplementation for deficiency could possibly benefit patients with chronic liver disease, compared with the controls in the same region [18,19,20].

Therefore, we systematically reviewed and meta-analyzed the evidence regarding the body selenium status and selenium intake in the risk of developing chronic liver diseases. To achieve this aim, we included both observational studies and interventional trials, retrospective and prospective. We present the findings of analyses stratified by disease severity.

## 2. Methods

### 2.1. Data Sources and Search Strategy

This meta-analysis was performed and written in accordance with the PRISMA standard guidelines and checklists [21]. The study protocol was registered in the International Prospective Register of Systematic Reviews (PROSPERO) database (registration number: CRD42020210144; https://www.crd.york.ac.uk/prospero/; accessed on 2 February 2020). PubMed, EMBASE, Cochrane Library, and Web of Science Core databases were searched from inception to April 2021 for observational studies and interventional trials that investigated body selenium status in patients with chronic liver diseases. In addition, the gray literature was manually searched in Google Scholar and Google databases. To minimize publication bias and avoid missing any relevant studies, reference lists and review papers on this topic were also reviewed. The search used a combination of keywords, including (selen* or selenium or selenium (MesH)) and (liver (MesH) or hepat* or liver disease). No language restrictions were set on the literature searches and retrieval. A flow diagram outlining the systematic review process is provided in Appendix A.

### 2.2. Inclusion and Exclusion Criteria

The following inclusion criteria were used to select articles for the meta-analysis: (1) selenium levels in whole blood, serum, plasma, toenails, hair, urine, feces, or liver tissue had been measured in the study; (2) mild, moderate, or severe chronic liver diseases such as fatty liver diseases, hepatitis, cirrhosis, and cancer were included; (3) main outcomes were chronic liver diseases incidence or/and mortality; (4) the number of cases and controls, mean and standard deviation (SD) for both groups, estimated odds ratio (OR) or hazard ratio (HR) with corresponding 95% confidence intervals (CI) for cases versus controls. We excluded narrative reviews, meeting abstracts, letters, case reports, conference papers, laboratory studies, and studies lacking data relevant to the association between body selenium status and chronic liver diseases. When several publications reported on the same study, we selected only the study with the most patients.

### 2.3. Data Extraction and Quality Assessment

Two researchers (YDL and TL) independently screened the titles, abstracts, and full-text articles to identify potentially relevant articles. In case of any disagreement over the eligibility for any study between the two reviewers, all these dissensions were thoroughly discussed, and a full-text assessment was conducted accordingly. Two investigators (YDL and BBX) independently extracted the following data from each eligible study in a standardized Microsoft Excel sheet, including the name of the first author, study design, year of publication, study region, study design, the country in which the study was conducted, the types of liver diseases, sample sources, selenium measurement, selenium doses, sample size, HR, or OR with 95% CIs, and adjusted confounders. If the study reported several multivariable-adjusted effect estimates, the effect estimates were selected and maximally adjusted for potential confounders, including age, sex, aspartate transaminase (AST), alanine transaminase (ALT), sarcopenia, nutrition, and alcohol usage. The Newcastle Ottawa Scale (NOS) was used to assess the quality of the cohort and case-control studies; the Jadad scale was used in cross-sectional studies, and the Agency for Healthcare Research and Quality (AHRQ) scale was used in RCT studies. NOS scores ≥ 7, Jadad scales ≥ 7, and AHRQ scale ≥ 4 are regarded as high methodological quality [22,23,24].

### 2.4. Statistical Analysis

Standard mean difference (SMD) and 95% CIs were pooled to evaluate the association between selenium levels and chronic liver diseases severity. SMD was used when studies reported different units or scales for the outcome [23]. OR and HR were pooled as relative risks (RR), which was used as the common measure of association across studies. Predominantly, HR, OR, or incidence rate ratios (IRR) can be directly considered as RR, because these estimates are approximate when event rates are low [24]. Moreover, subgroup analysis was first conducted by the types of chronic liver diseases, and further stratified by study regions, sample sources of body selenium, study design, and year of publication. Although selenium status varies widely in different parts of the world, the current reference level of selenium in the blood is fixed at 70 to 150 µg/L [25]. According to the mean baseline of blood selenium level in the normal control group, we roughly divided studies into three major groups (<70, 70–150, >150 µg/L) and carried out a subgroup analysis. Sensitivity analysis was applied to estimate the influence of each individual study or a group of studies on the pooled RR. A priori, a random-effects model was applied to calculate SMD and pooled RR with 95% CIs due to the anticipated clinical and methodological heterogeneity, which was considered more conservative than fixed-effect models, as it can explain the heterogeneity within and between studies. Statistical heterogeneity was evaluated by using the *Q* statistics and *I*^2^ statistics [26]. The *I*^2^ statistic indicated the percentage of overall variation across the studies due to heterogeneity rather than chance, and the heterogeneity was defined as low if *I*^2^ < 30%, moderate if 30% ≤ *I*^2^ ≤ 50%, and high if *I*^2^ ≥ 50%, as was described previously [27]. The random effects of the calculation of summary effect measures were used if *I*^2^ ≥ 50%. Conversely, the fixed effect was used for summary effect measures estimation if *I*^2^ ≥ 50% [28]. In subgroup analysis, we classified fatty liver diseases into alcoholic fatty liver disease, NAFLD, and simple fatty liver disease; hepatitis into viral hepatitis (HBV, HCV, active viral hepatitis, and persist hepatitis), alcoholic hepatitis, and other types of chronic hepatitis; cirrhosis was divided into primary cirrhosis, alcoholic cirrhosis, and other types of cirrhosis (primary biliary cirrhosis and cryptogenic cirrhosis, etc.); and liver cancer (hepatocellular carcinoma, biliary tract cancer, and intrahepatic bile duct cancer). The possibility of publication bias was assessed by using a combination of Begg’s test and Egger’s test and funnel plots [29]. Meta-regression was performed to assess the potential dose-response relationship between the increment in selenium level and chronic liver diseases incidence. All analyses were performed using STATA version 15.0 (StataCorp LP, College Station, TX, USA). *p*-value < 0.05 was considered statistically significant unless otherwise noted.

## 3. Results

### 3.1. Study Characteristics

A total of 2223 potentially relevant articles were initially identified with the searched databases, and 370 records were excluded because of duplication. After screening the titles and abstract reviews, 1715 articles were further excluded, resulting in a full review of 138 articles. Finally, only 50 articles met the inclusion criteria (see Figure 1), including 4 cohort studies, 35 case-control studies, 1 nested case-control study, 6 cross-sectional studies, and 4 randomized controlled triads. Appendix A gives an overview of the characteristics of the extracted studies. Included studies were conducted in Asia (19 studies), Africa (1 study), Europe (25 studies), and the USA (5 studies). On the other hand, according to the types of liver diseases, 7 were on fatty liver diseases [16,17,30,31,32,33,34], 22 on hepatitis [13,19,33,35,36,37,38,39,40,41,42,43,44,45,46,47,48,49,50,51,52,53], 32 on liver cirrhosis [13,30,31,33,35,36,37,38,39,41,42,44,48,50,52,54,55,56,57,58,59,60,61,62,63,64,65,66,67,68,69,70], and 15 studies on liver cancer [13,35,40,44,48,50,59,64,65,71,72,73,74,75,76]. In total, there were 9875 cases (comprising 5621 cases of fatty liver diseases, 1635 cases of hepatitis, 1455 cases of liver cirrhosis, and 1164 cases of liver cancer) and 12,975 controls. In addition, four articles also calculated the liver disease risk ratios (e.g., OR, RR, or HR and 95% CIs). Most of the studies cited here measured blood selenium (*n* = 45), three studies measured hair selenium level, three studies measured liver selenium, and the remaining two studies measured nail selenium level. No eligible studies were measuring selenium in urine or feces. Overall, 137 doses of body selenium were extracted (48 doses in whole blood samples, 53 doses in serum samples, 17 doses in plasma samples, 13 doses in hair samples, 4 doses in liver samples, and 2 doses in toenail samples).

### 3.2. Association between Selenium Level and Chronic Liver Diseases

First, we included all the individuals to calculate the average selenium status in patients and healthy controls using random-effects models. We found an adverse association between body selenium status and chronic liver diseases (SMD = −1.70, 95% CI: −2.30 to −1.11, *n* = 137), but the heterogeneity was high (*I*^2^ = 99.5%, *p* < 0.001) (see Appendix A). Due to discrepancies in the literature, we performed subgroup analysis based on the severity of chronic liver diseases. We found that fatty liver disease patients had an equivalent level of selenium to healthy controls (SMD = 1.06, 95% CI: −1.78 to 3.89, *n* = 11) (see Appendix A), whereas patients with hepatitis (SMD = −1.78, 95% CI: −2.22 to −1.34, *n* = 44), liver cirrhosis (SMD = −2.06, 95% CI: −2.48 to −1.63, *n* = 57), and liver cancer (SMD = −2.71, 95% CI: −3.31 to −2.11, *n* = 25) had a lower selenium level than healthy controls, regardless of the baseline of selenium level in the body (see Appendix A). The funnel plot of hepatitis, liver cirrhosis, and liver cancer articles all appeared symmetric, and egger’s test detected no publication bias. Although the funnel plot of fatty liver diseases showed bias, no statistically significant publication bias was found by Egger’s test. There was no presence of publication bias for the studies (Appendix A).

Additionally, we performed a further subgroup analysis according to the types of diseases, sample sources, different regions, years of publication, and the reference blood selenium levels (optimal or suboptimal) (see Table 1, Appendix A).

For fatty liver diseases, stratified analyses by types of diseases showed that, compared with the controls, alcoholic fatty liver disease patients (SMD = −1.29, 95% CI: −2.08 to −0.50, *n* = 3) had significantly lower selenium levels, while simple fatty liver disease patients had no difference (SMD = −0.51, 95% CI: −0.90 to −0.12, *n* = 4) and NAFLD patients had a higher level (SMD = 4.39, 95% CI: −0.55 to 9.34, *n* = 4), although not significantly (see Appendix A). Stratified analyses by study regions showed that fatty liver patients had a significantly lower selenium level in Europe than controls, but no difference in the USA and a higher level in Asia (see Appendix A). Stratified analyses by sample sources showed that fatty liver disease patients had a significantly lower hair selenium level than controls, but no difference in whole blood, serum, and plasma (see Appendix A). Stratified analyses by study design showed that fatty liver disease patients had a significantly lower selenium level than controls in case-control studies, but no difference in cross-sectional studies (see Appendix A). Stratified analyses by year of publication showed that fatty liver disease patients had a significantly lower selenium level than controls in studies carried out before 1990, but no difference in studies carried out after 1991 (see Appendix A). Subgroup analysis stratified by the mean baseline blood selenium level in the normal control group showed that, interestingly, when in an optimal selenium level (70–150 µg/L), fatty liver disease patients (3 in alcohol liver disease and 1 in NAFLD) had a significantly lower blood selenium level than controls (SMD = −0.92, 95% CI: −1.87 to 0.02, *n* = 4), but in a suboptimal selenium level (>150 µg/L), there was no significant difference between NAFLD patients and controls (SMD = 5.84, 95% CI: −3.29 to 14.97, *n* = 3) (see Appendix A).

For hepatitis, stratified analyses by types of diseases showed that, compared with controls, viral hepatitis patients (SMD = −1.88, 95% CI: −2.42 to −1.35, *n* = 34) and alcoholic hepatitis patients (SMD = −1.36, 95% CI: −1.94 to −0.77, *n* = 9) had a significantly lower selenium level (see Appendix A). Stratified analyses by study regions showed that hepatitis patients had a significantly lower selenium level in Europe and Asia than controls, while no difference was found in Africa (see Appendix A). Further subgroup analysis stratified by sample sources showed that hepatitis patients had a significantly lower selenium level than controls in each sample (see Appendix A). Subgroup analysis stratified by study design showed that hepatitis patients had a significantly lower selenium level than controls in each type of study (see Appendix A). Subgroup analysis stratified by year of publication showed that hepatitis patients had a significantly lower selenium level than controls in most studies, except for in studies carried out from 1991 to 2000 (see Appendix A). Subgroup analysis stratified by the mean baseline blood selenium level in the normal control group showed that whether in an optimal (70–150 µg/L (SMD = −1.22, 95% CI: −1.62 to −0.82, *n* = 24)) or suboptimal (<70 µg/L (SMD = −2.38, 95% CI: −3.57 to −1.19, *n* = 4) or >150 µg/L (SMD = −2.65, 95% CI: −4.04 to −1.26, *n* = 12)) selenium level, hepatitis patients had a significantly lower blood selenium level than controls (see Appendix A).

For liver cirrhosis, stratified analyses by types of diseases showed that alcoholic cirrhosis patients (SMD = −2.45, 95% CI: −2.99 to −1.90, *n* = 21) and other cirrhosis patients (SMD = −2.41, 95% CI: −2.94 to −1.89, *n* = 27) had a significantly lower selenium level than controls, but no difference in primary liver cirrhosis (SMD = 0.90, 95% CI: −1.98 to 3.79, *n* = 9) (see Appendix A). Stratified analyses by study regions showed that liver cirrhosis patients had a significantly lower selenium level in Europe, the USA, and Asia than controls (see Appendix A). Stratified analyses by sample sources showed that liver cirrhosis patients had a significantly lower selenium level than controls in most samples, except for the liver (see Appendix A). Subgroup analysis stratified by study design showed that liver cirrhosis patients had a significantly lower selenium level than controls in each type of study (see Appendix A). Stratified analyses by year of publication showed that liver cirrhosis patients had a significantly lower selenium level than controls in most studies, except for in studies carried out before 1990 (see Appendix A). Subgroup analysis stratified by the mean baseline blood selenium level in the normal control group showed that whether in an optimal (70–150 µg/L) or suboptimal (<70 µg/L or >150 µg/L) selenium level, liver cirrhosis patients had a significantly lower blood selenium level than controls (see Appendix A).

For liver cancer, almost all studies of liver cancer have only included patients with hepatocellular carcinoma, so we did not perform subgroup analysis by the types of liver cancer. Stratified analyses by study regions showed that liver cancer patients had a significantly lower selenium level in Europe, the USA, and Asia than controls (see Appendix A). Subgroup analysis stratified by sample sources showed that liver cancer patients had a significantly lower selenium level than controls in each sample (see Appendix A). Subgroup analysis stratified by study design showed that liver cancer patients had a significantly lower selenium level than controls in each type of study (see Appendix A). Subgroup analysis stratified by year of publication showed that liver cancer patients had a significantly lower selenium level than controls in each publication year (see Appendix A). Subgroup analysis stratified by the mean baseline blood selenium level in the normal control group showed that whether in an optimal (70–150 µg/L) or suboptimal (<70 µg/L or >150 µg/L) selenium level, liver cancer patients had a significantly lower blood selenium level than controls (see Appendix A).

### 3.3. Association between Body Selenium Status and Incidence of Advanced Chronic Liver Diseases

The multivariable-adjusted RRs for each study and comparison of the combined RR between the highest and lowest selenium levels are shown in Figure 2. A random-effects model was used to calculate the overall effect size. The incidence of advanced chronic liver diseases was significantly reduced in the individuals with high body selenium status compared to those with low baseline selenium status (RR = 0.59, 95% CI: 0.49 to 0.72, *p* = 0.415). No significant heterogeneity (*I*^2^ = 0.1%) was found. Begg’s test detected no publication bias (*p* = 0.452). Furthermore, we conducted a meta-regression analysis and revealed a statistically significant linear dose-response relationship between blood selenium level and chronic liver diseases incidence. However, no statistically significant dose-response relationship was found between blood selenium increment and the risk for chronic liver diseases incidence (RR = 1.01, 95% CI: 0.78 to 1.30, *p* = 0.663).

### 3.4. Association between Selenium Intake and Chronic Liver Diseases Risk

The result of multivariable-adjusted RRs for the association between selenium intake and risk of chronic liver diseases is shown in Figure 3. A tolerable upper intake level of selenium is 400 μg/day [40,77]. We found that within this safe dose range, as selenium intake or body status increased, the potential risk of advanced chronic liver diseases decreased. When selenium intake was high but below 400 μg/day, the risk of hepatitis (RR = 0.64, 95% CI: 0.50 to 0.81, *n* = 4; *I*^2^ = 0%, *p* = 0.505) and liver cancer (RR = 0.61, 95% CI: 0.50 to 0.75, *n* = 9; *I*^2^ = 26.4%, *p* = 0.209) reduced significantly without significant heterogeneity. On the contrary, increased selenium intake was associated with increased risk of NAFLD (RR = 1.60, 95% CI: 1.13 to 2.25, *n* = 2), but it had significant heterogeneity (*I*^2^ = 96.1%, *p* < 0.001). The funnel plot of included articles appears asymmetric (see Appendix A).

## 4. Discussion

This study systematically reviewed and meta-analyzed the evidence regarding the relationship between the body selenium levels and the risk of chronic liver diseases. To achieve this aim, both observational and interventional studies, retrospective and prospective, were included. Finally, 50 relevant articles with 9875 cases and 12,975 controls in total were involved. The meta-analysis suggested an adverse association between body selenium status and chronic liver diseases.

Specifically, subgroup analysis by the severity of liver disease indicated that patients with advanced chronic liver diseases (hepatitis, liver cirrhosis, and liver cancer) had a significantly lower selenium level than healthy controls overall, whether the mean baseline of body selenium levels of controls was optimal or suboptimal. Concomitantly, as the liver disease developed, the negative correlation was more pronounced, whereas the relationship was controversial in the early stages of the disease. The selenium level was negatively correlated with alcoholic fatty liver disease, while no remarkable difference was observed in simple fatty liver disease, and in contrast, the relationship turned out to be positive in NAFLD. In addition, we also found that the whole blood selenium concentration of fatty liver diseases patients was higher than that of normal people, but the opposite phenomenon was observed in other sample sources. Studies have shown that toenail clippings are considered a superior marker of selenium status because they provide comprehensive measurements over a period of up to a year, while blood levels are considered a more appropriate short-term marker of selenium exposure [84,85]. However, few studies have focused on selenium status in the nails and hair among fatty liver disease patients, and the data of NAFLD are especially lacking. Nevertheless, this aspect needs further investigation.

The liver directly acquires selenium from dietary form (primarily selenomethionine) and converts the dietary form into selenocysteine by transsulfuration, which is used to form various selenoproteins and perform the functions of selenium [86]. SELENOP, a secreted glycoprotein, is mainly produced by the liver and secreted into the circulation, which controls selenium transport and storage, playing a key role in selenium homeostasis and the development of chronic liver disease [87]. SELENOP has been shown to protect against liver injury, hepatocyte necrosis, and apoptosis in both oxidative stress and inflammatory-related mechanisms. Moreover, SELENOP has been recently implicated in the regulation of glucose metabolism and insulin sensitivity [88]. Insufficient dietary selenium intake can lead to a low body selenium level. In addition, liver dysfunction may also affect the conversion of selenium to selenite from selenomethionine, leading to more serious functional selenium deficiency in patients [89]. The current meta-analysis indicated that a physiologically higher baseline selenium level may be a protective factor for the incidence of advanced chronic liver diseases, especially for the end-stage, which was consistent with the previous meta-analysis of selenium for preventing cancer [90]. Among all the study designs of the multivariate-adjusted RR, there was a statistically significant decrease in advanced chronic liver diseases incidence overall. Similarly, Li et al. found that supplementing participants with selenium for 3 years can successfully reduce the incidence of liver cancer [91]. Furthermore, we found that selenium intake within the tolerable range (<400 µg/day) was negatively associated with advanced chronic liver diseases; however, it was positively associated with the prevalence of NAFLD. In advanced chronic liver diseases, it seems that selenium deficiency is largely due to liver damage. Our results agree with previous reports, which suggest that appropriate selenium supplementation significantly increases blood selenium levels and helps control the progression of advanced chronic liver diseases. In addition, we found that NAFLD patients had a higher (although not significantly) selenium level than healthy controls, whether the mean baseline blood selenium level of controls was optimal (70–150 µg/L) or suboptimal (>150 µg/L). In other words, if selenium intake is adequate, the blood selenium would not drop but even increase in the early stage of NAFLD. Currently, NAFLD studies of body selenium levels are scarce and inconsistent. Several studies have shown higher or similar SELENOP levels in patients and controls [30]. Moreover, another study showed that selenium supplementation in replete individuals did not cause an increase in plasma glutathione peroxidase activity or SELENOP concentration [77]. Accumulating studies implied that the physicochemical and biological properties of selenium differ substantially depending on its valence: only selenite Se4+ but not Se6+ can act as an oxidant in the redox reactions [1,2]. Some organic selenium compounds also possess antioxidant properties after transformation to inorganic forms of selenium in vivo [1,2]. Specifically, selenite can disrupt parafibrin (a protective protein coat of cancer cells) formation by oxidizing sulfhydryls to disulfides in fibrinogen, thus potentially increasing tumor-immune recognition and eliminating cancer cells by inducing apoptosis. Importantly, selenite can also directly activate the natural killer (NK) cells [92]. These results together with our findings may indicate that once the selenium requirement has been met, the synthesis of selenoproteins in the liver would not be affected in patients with NAFLD, blood selenium levels might not mirror selenium intake, and the chemical forms of selenium in dietary or supplement form can affect the biosynthesis of selenoproteins [1,2]. Thus, the optimal dose of selenium supplementation for preventing and treating chronic liver diseases in patients of different stages is still unclear. However, in view of these properties of selenium, keeping selenium supplementation in a suitable chemical form and within a safe dosage (<400 µg/day) can benefit patients of severe chronic liver diseases (hepatitis, liver cirrhosis, and liver cancer).

It is a bit complicated in the case of fatty liver disease. By analyzing the overall result of fatty liver disease, a positive correlation with high heterogeneity can be found between the selenium level and fatty liver disease. Nonetheless, the selenium level was negatively correlated with simple fatty liver disease and alcoholic fatty liver disease. When we excluded the studies of NAFLD, the heterogeneity of fatty liver disease reduced to 85.8%, suggesting the studies of NAFLD were the main source of heterogeneity (see Appendix A). Until now, the association between NAFLD and body selenium is inconsistent. Such uncertainty was probably derived from the imprudent diagnostic methods and the influence of the large sample size on the overall result. Currently, the diagnostic criteria for NAFLD have been variously defined. Liver biopsy is the current gold standard for diagnosis of NAFLD, followed by magnetic resonance imaging (MRI). Polyzos et al. observed that the SEPP levels were lower in patients with biopsy-proven NAFLD compared with the controls [93]. However, three studies [17,32,34] that used B-mode ultrasound for NAFLD diagnosis showed opposite results. Although B-mode ultrasound examination is a convenient way, its sensitivity (91%) and specificity (89%) were significantly decreased compared to the gold standard. In addition, it did not perform well in morbidly obese patients and may result in misdiagnosis (because of insensitivity) if the steatosis is ≤30%. The diagnosis of NAFLD in the included studies was a B-mode ultrasound, which may have led to an overestimation of screening benefits [94]. Such imprudent diagnoses could not only lead to a misclassification that biased the final result but also be one of the reasons why the correlations of the selenium level and fatty liver disease are contradictory between Asia and other continents. The underlying mechanism of selenium in the pathogenesis of NAFLD is not yet fully elucidated. Some studies have found that a high circulating selenium level is correlated with impaired insulin signaling and could potentially modulate liver insulin resistance, and insulin resistance plays a pivotal role in the development of hepatic lipid accumulation [85]. Further larger sample size studies with a definite diagnosis of NAFLD are needed to elucidate the relationship between the selenium level and NAFLD.

The information of body selenium status can be reflected by multiple biological activities of selenium in different sample sources. In epidemiological studies, blood and toenails have been frequently used to assess selenium status [95]. The blood selenium levels (whole blood, serum, or plasma) indicated the portion of ingested selenium that was absorbed and retained [96]. Nail and hair samples represented past selenium status [97], and the liver tissue sample indicated a portion of retained selenium that may become available for functional purposes over the medium-long term [96]. A previous meta-analysis by Gong et al. reported that selenium concentrations in toenails, whole blood, and serum were all inversely associated with the risk for HCC [98]. In this meta-analysis, there were six different biological samples used to assess selenium status, including whole blood, serum, plasma, toenails, hair, and liver tissue. Subgroup analysis by sample sources found that most of the advanced chronic liver diseases were inversely associated with selenium levels. However, in patients with cirrhosis, selenium in tissues and nails was positively associated with liver disease, possibly because the liver samples were from children. The etiology of cirrhosis in children is different from that in adults. Olave et al. indicated that the major cause of cirrhosis in children (patients < 18 years old) was congenital cholestatic syndromes, genetic-metabolic disorders, and autoimmune disorders [99].

### Strengths and Limitations

One of the main advantages of our study is that it covers a wide range of chronic liver diseases from fatty liver disease to liver cancer, a large sample size of former credible research using different researching methods, and is conducted in multiple countries including 21 countries throughout 4 continents (Asia, Africa, Europe, and North America). Unlike prior studies, selenium was measured from multiple body parts including whole blood, serum, plasma, liver, hair, and toenail, instead of a dietary assessment questionnaire, which can not only avoid recall bias but also be more precise on selenium measurement and indicate short-term/long-term absorption and storage of selenium. Although a statistically prominent association between body selenium and liver diseases was found, the heterogeneity in certain groups was still an obstacle to a comprehensive conclusion. To prudently find out the origin of heterogeneity and explicitly explain the inconsistency of the associations between selenium and fatty liver disease in former studies, sub-group analysis or sensitivity analysis was meticulously performed. However, the heterogeneity of the association between NAFLD and body selenium level was still a bit high. The possible reasons, as were mentioned in the discussion before, could be the misclassification caused by the imprudent diagnosis. The results from different samples were, however, contradictory and further led to a massive influence on the overall result when considering the weight of this study. It is worth noting that dietary changes in patients with chronic liver disease may affect selenium levels in their bodies, but we could not eliminate this impact. As we used secondary data for analysis, omitted variables could bias our results. Thus, the findings should be acknowledged with caution.

As selenium concentrations were only measured at baseline, and concentrations may have changed during the study period, this time-varying confusion may skew the association in an unknown direction. We should also recognize that some studies have increased weight in pooled SMD, appearing twice due to different selenium concentrations in different clinical subgroups. Therefore, the association between selenium and chronic liver disease may be exaggerated by these studies. We only analyzed the association between selenium concentrations at different sources and chronic liver disease, so we do not know enough about whether selenoprotein levels or other forms of selenium protection are sufficient to cause increased susceptibility to liver disease. Furthermore, although we conducted subgroup analysis by study design, sample type, disease, year of publication, study region, and the reference blood selenium level, the source of heterogeneity remains largely undetected. Thus, we must interpret the results with caution.

## 5. Conclusions

In conclusion, our work demonstrated that the body selenium status decreased significantly in patients with advanced chronic liver diseases than the controls, and a physiologically high blood selenium level was associated with decreased advanced chronic liver diseases incidence; however, such an association was inconsistent in fatty liver diseases, especially in NAFLD. More importantly, high selenium intake within a safe dosage (<400 µg/day) tended to be a protective factor in advanced chronic liver diseases, but such evidence for fatty liver diseases was not enough. Further research with larger RCTs is warranted to confirm these associations in NAFLD. Additionally, further studies involving the relationship between parameters affecting the risk of chronic liver disease (e.g., morbidity and mortality, and progression to decompensation in cirrhosis) and body selenium levels can confirm this conclusion, as the results of the dose–response relationship between blood selenium increment and parameters affecting the risk of chronic liver diseases in this meta-analysis remain not statistically significant.

## Figures and Tables

**Figure 1 nutrients-14-00952-f001:**
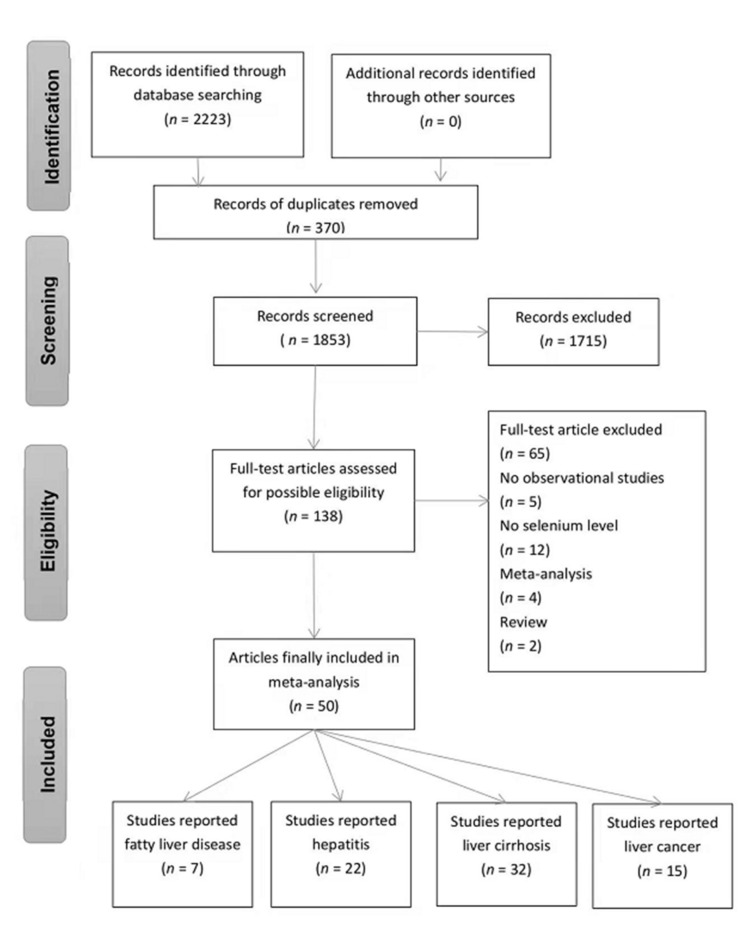
PRISMA diagram of primary studies.

**Figure 2 nutrients-14-00952-f002:**
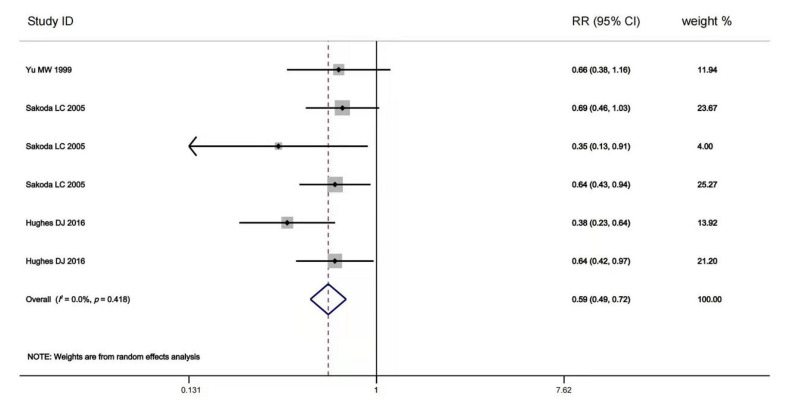
Forest plot of estimates comparing the elevated risk of advanced chronic liver diseases for the highest baseline selenium level compared to the lowest baseline selenium level [40,74,76]. The width of the black line represents 95% CI, which were obtained from a linear meta-regression without a constant using logarithmic RR for advanced chronic liver diseases as the dependent variable and the difference in selenium level to the study-specific reference group as the independent variable (either the highest baseline selenium level or the lowest baseline selenium level). The random-effects accounted for clustered variance in each study.

**Figure 3 nutrients-14-00952-f003:**
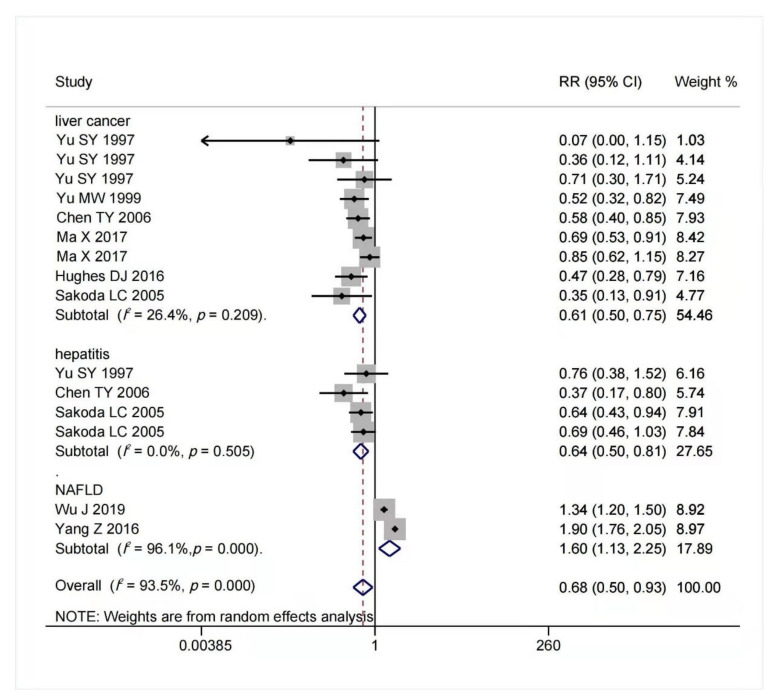
Pooled RRs of selenium intake and chronic liver diseases. Stratified by disease progression [17,40,73,75,78,79,80,81,82,83].

**Table 1 nutrients-14-00952-t001:** SMD by study design, region, sample sources, year of publication, and blood selenium levels.

Outcome	Group Variable	Subgroup	Doses, *n*	SMD (95% CI)	*I* ^2^	*p* of *I*^2^
Fatty liver diseases	Overall		11	1.06 (−1.78, 3.89)	99.9%	<0.001
	Disease types	Alcoholic fatty liver diseases	3	−1.29 (−2.08, −0.50)	71.7%	0.032
		Simple fatty liver diseases	4	−0.51(−0.90, −0.12)	0	0.864
		NAFLD	4	4.39(−0.55, 9.34)	100%	<0.001
	Study design	Case-control	7	−0.86 (−1.29, −0.43)	55.1%	0.038
		Cross-sectional	4	4.39 (−0.55, 9.34)	100%	<0.001
	Study regions	Asia	7	2.21 (−3.39, 7.81)	100%	<0.001
		Europe	3	−1.29 (−2.08, −0.50)	71.1%	0.032
		USA	1	0.06 (−0.01, 0.12)		
	Sample sources	Hair	4	−0.51 (−0.90, −0.12)	0%	0.864
		Serum	3	−0.98 (−2.13, 0.17)	95.4%	<0.001
		Whole blood	3	5.84 (−3.29, 14.97)	100%	<0.001
		Plasma	1	−0.76 (−1.65, 0.13)		
	Year of publication	1973–1990	2	−1.55 (−2.76, −0.33)	83.1%	0.015
		1991–2000	1	−0.76 (−1.65, 0.13)		
		2011–now	8	1.95 (−1.46, 5.36)	100%	<0.001
	Blood selenium level ^1^	70–150 µg/L	4	−0.92 (−1.87, 0.02)	93.5%	<0.001
		>150 µg/L	3	5.84 (−3.29, 14.97)	100%	<0.001
Hepatitis	Overall		44	−1.78 (−2.22, −1.34)	95.8%	<0.001
	Disease types	Other hepatitis	1	−2.42 (−3.69, −1.16)		
		Viral hepatitis	34	−1.88 (−2.42, −1.35)	96.6%	<0.001
		Alcoholic hepatitis	9	−1.36 (−1.94, −0.77)	84.6%	<0.001
	Study design	Case-control	36	−1.07 (−1.31, −0.84)	78.1%	<0.001
		Cross-sectional	1	−0.89 (−1.27, −0.52)		
		Cohort	4	−6.67 (−9.98, −3.37)	98.9%	<0.001
		RCT	3	−6.03 (−10.18, −1.87)	99.1%	<0.001
	Study regions	Asia	20	−2.69 (−3.55, −1.84)	97.9%	<0.001
		Europe	20	−1.17 (−1.51, −0.83)	81.0%	<0.001
		USA	1	−2.42 (−3.69, −1.16)		
		Africa	3	−0.55 (−0.92, 0.19)	0%	0.885
	Sample sources	Hair	4	−1.27 (−1.69, −0.84)	28.1%	0.244
		Serum	15	−1.46 (−2.06, −0.85)	92.3%	<0.001
		Whole blood	18	−2.08 (−2.85, −1.31)	97.3%	<0.001
		Plasma	7	−2.16 (−3.51, −0.82)	95.6%	<0.001
	Year of publication	1973–1990	7	−1.44 (−2.16, −0.72)	80.6%	<0.001
		1991–2000	5	−0.44 (−1.01, 0.13)	77.3%	0.001
		2001–2010	16	−1.80 (−2.45, −1.16)	94.7%	<0.001
		2011–now	16	−2.21 (−3.16, −1.25)	97.4%	<0.001
	Blood selenium level ^1^	<70 µg/L	4	−2.38 (−3.57, −1.19)	95.3%	<0.001
		70–150 µg/L	24	−1.22 (−1.62, −0.82)	90.4%	<0.001
		>150 µg/L	12	−2.65 (−4.04, −1.26)	98.1%	<0.001
Liver cirrhosis	Overall		57	−2.06 (−2.48, −1.63)	95.4%	<0.001
	Disease types	Alcoholic cirrhosis	21	−2.45 (−2.99, −1.90)	92.0%	<0.001
		Other cirrhosis	21	−2.41 (−2.94, −1.69)	95.0%	<0.001
		Primary cirrhosis	9	0.90 (−1.98, 3.79)	98.2%	<0.001
	Study design	Case-control	46	−1.66 (−2.09, −1.23)	94.5%	<0.001
		Cross-sectional	2	−2.37 (−4.50, −0.25)	96.7%	<0.001
		Cohort	1	−1.67 (−2.58, 0.76)		
		RCT	8	−4.51 (−6.12, −2.90)	97.1%	<0.001
	Study regions	Asia	15	−2.12 (−3.30, −0.94)	97.9%	<0.001
		Europe	38	−1.95 (−2.37, −1.53)	93.1%	<0.001
		USA	4	−2.86 (−3.65, −2.07)	53.3%	0.093
	Sample sources	Liver	4	3.03 (−4.96, 11.03)	99.2%	<0.001
		Serum	28	−2.18 (−2.57, −1.80)	90.9%	<0.001
		Whole blood	13	−3.39 (−4.56, −2.22)	97.1%	<0.001
		Plasma	8	−2.08 (−2.65, −1.51)	68.9%	0.002
		Nail	1	6.63 (5.01, 8.24)		
		Hair	3	−1.68 (−2.22, −1.14)	50.1%	0.135
	Year of publication	1973–1990	17	−0.77 (−1.98, 0.44)	96.9%	<0.001
		1991–2000	16	−1.89 (−2.34, −1.44)	85.5%	<0.001
		2001–2010	9	−2.21 (−3.00, −1.41)	94.3%	<0.001
		2011–now	15	−3.15 (−4.09, −2.21)	96.6%	<0.001
	Blood selenium level ^1^	<70 µg/L	5	−2.09 (−2.62, −1.55)	81.8%	<0.001
		70–150 µg/L	37	−2.00 (−2.35, −1.64)	91.0%	<0.001
		>150 µg/L	7	−5.44 (−7.77, −3.11)	95.6%	<0.001
Liver cancer	Overall		25	−2.71 (−3.31, −2.11)	97.5%	<0.001
	Study design	Case-control	12	−1.64 (−2.23, −1.06)	91.7%	<0.001
		Nest case-control	1	−0.45 (−0.63, −0.26)		
		Cohort	4	−0.74 (−1.37, −0.11)	93.7%	<0.001
		RCT	8	−6.15 (−8.10, −4.20)	98%	<0.001
	Study regions	Asia	14	−3.92 (−4.91, −2.93)	98.4%	<0.001
		Europe	10	−1.22 (−1.81, −0.63)	92.5%	<0.001
		USA	1	−2.36 (−3.09, −1.63)		
	Sample sources	Hair	2	−2.31 (−3.93, −0.69)	93.9%	<0.001
		Serum	8	−2.38 (−3.37, −1.40)	95.7%	<0.001
		Whole blood	13	−3.29 (−4.26, −2.32)	98.0%	<0.001
		Plasma	1	−2.36 (−3.53, −1.19)		
		Nail	1	−0.45 (−0.63, −0.26)		
	Year of publication	1973–1990	2	−2.08 (−2.79, −1.38)	24%	0.251
		1991–2000	4	−1.00 (−1.80, −0.19)	91.7%	<0.001
		2001–2010	5	−1.27 (−2.24, −0.30)	95.2%	<0.001
		2011–now	14	−3.98 (−4.98, −2.97)	98.1%	
	Blood selenium level ^1^	<70 µg/L	2	−2.96 (−3.43, −2.50)	4.2%	0.307
		70–150 µg/L	11	−1.58 (−2.23, −0.93)	95%	<0.001
		>150 µg/L	9	−4.47 (−6.43, −3.11)	98.5%	<0.001

RCT: randomized controlled trial; SMD: standard mean differences; 95% CI: 95% confidence interval; ^1^ blood selenium level includes whole blood, serum, and plasma.

## Data Availability

The data presented in this study are available in the article, Appendix A, or upon request to the authors.

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
