# Peer review of "Selenium Status in Patients with Chronic Liver Disease: A Systematic Review and Meta-Analysis"

_nutrients, 2022, doi:10.3390/nu14050952_

Round 1
Reviewer 1 Report
My comments were only partially addressed. I still doubt the relevance of the findings. However, the meta-analysis was done properly - it's simply the original data that are inconclusive and inconsistent.
The authors did not provide sufficient responses to my comments.
- How are selenium levels associated with outcome parameters (e.g. risk of decompensation in cirrhosis, mortality etc)?
- How do selenium levels change longitudinally in patients - and would this substantiate the authors' revised conclusion on selenium supplementation?
- If the authors cannot control for confounders (sacropenia, nutrition, alcohol etc), this needs to be stated as a major limitation and clearly disclosed to the readers.
Reviewer 2 Report
The authors present selenium as an element very poorly. I have not noticed that the authors specify the chemical forms of selenium and its impact on human health.
There is no information about the pathological conditions that the concentration of selenium can cause.
There is no information about the possible formation of protein amyloidosis. What is the significance of selenium in this matter and its redox form?
Why did the authors not pay attention to the scope of application of selenium and its reducing and oxidative properties. It all depends on certain situations. The authors of these issues do not discuss at all.
What are the nutritional recommendations for acute and chronic hepatitis in patients? Should the diet be rich in selenium? The authors should specify this.
Add publications that will help you explain this problem:
(2017). Application of sodium selenite in the prevention and treatment of cancers. Cells, 6 (4), 39.
(2019). Redox-active selenium in health and disease: a conceptual review. Mini Reviews in Medicinal Chemistry, 19 (9), 720-726.
(2018). Pathophysiological significance of protein hydrophobic interactions: An emerging hypothesis. Medical Hypotheses, 110, 15-22.
The concentration of selenium in the blood plasma in patients with chronic liver diseases should be characterized by a low content of selenium compared to the reference values. In healthy people, the range of serum selenium concentrations
blood count should be 0.8 - 2 µmol / L. Do the authors agree with this? Please provide information in the manuscript.
The research methods used are not supported by any literature. It should be absolutely corrected.
Conclusion is very short and does not cover the entire research problem that has been presented. The meta-analysis should be more elaborate and discussed. The authors should absolutely correct it
All these issues must be clarified.
Author Response
请参阅附件

Round 2
Reviewer 1 Report
The work is improved.
Reviewer 2 Report
The authors corrected the presented scientific article. They responded to every comment.
This manuscript is a resubmission of an earlier submission. The following is a list of the peer review reports and author responses from that submission.
Round 1
Reviewer 1 Report
In this article Lin et al., evaluated the association between liver disease and selenium status. The authors concluded that patients with hepatitis, liver cirrhosis and liver malignancy displayed significantly lower selenium levels than controls. On the contrary there was no significant difference in patients with fatty liver disease.
This is an extensive effort based on the appropriate statistical methodology that addresses an interesting topic with major clinical implications. However, prior to acceptance, the following issues should be addressed:
- A considerable shortening of the results section should be performed. Please focus on the pooled comparison results and provide a concise report of the subgroup findings
- Please highlight the exact screening algorithm that was used
- The term 'liver cancer' needs to be clarified (i.e. HCC).
Reviewer 2 Report
This article reports a meta-analysis on selenium in liver diseases. I have a couple of serious concerns.
1.In view of the very limited and heterogeneous data, the authors included all types of studies in this meta-analysis (e.g. selenium measures from different sources such as blood or urine). They also cover various stages of liver diseases. In the end, the data are not trustworthy. If anything, the authors should have performed a meta-analysis with all individual data, in order to identify potential confounders (e.g. by a multivariate analysis). It is currently impossible to draw meaningful results from the study, particularly from a clinical perspective.
2.Important confounders may be the nutritional status, sarcopenia and the etiology of liver disease (such as alcohol-related disease). This needs to be accounted and controlled for.
3.The potential benefit from an intervention targeting selenium is unclear.